# Presenilin 1 Modulates Acetylcholinesterase Trafficking and Maturation

**DOI:** 10.3390/ijms24021437

**Published:** 2023-01-11

**Authors:** María-Ángeles Cortés-Gómez, Víctor M. Barberá, Jordi Alom, Javier Sáez-Valero, María-Salud García-Ayllón

**Affiliations:** 1Unidad de Investigación, Hospital General Universitario de Elche, Fundación Para el Fomento de la Investigación Sanitaria y Biomédica de la Comunidad Valenciana (FISABIO), 03203 Elche, Spain; 2Centro de Investigación Biomédica en Red Sobre Enfermedades Neurodegenerativas (CIBERNED), 03550 Sant Joan d’Alacant, Spain; 3Instituto de Neurociencias de Alicante, Universidad Miguel Hernández-CSIC, 03550 Sant Joan d’Alacant, Spain; 4Unidad de Genética Molecular, Hospital General Universitario de Elche, 03203 Elche, Spain; 5Servicio de Neurología, Hospital General Universitario de Elche, Fundación Para el Fomento de la Investigación Sanitaria y Biomédica de la Comunidad Valenciana (FISABIO), 03203 Elche, Spain; 6Instituto de Investigación Sanitaria y Biomédica de Alicante (ISABIAL), 03010 Alicante, Spain

**Keywords:** Alzheimer’s disease, acetylcholinesterase, glycosylation, presenilin 1

## Abstract

In Alzheimer’s disease (AD), the reduction in acetylcholinesterase (AChE) enzymatic activity is not paralleled with changes in its protein levels, suggesting the presence of a considerable enzymatically inactive pool in the brain. In the present study, we validated previous findings, and, since inactive forms could result from post-translational modifications, we analyzed the glycosylation of AChE by lectin binding in brain samples from sporadic and familial AD (sAD and fAD). Most of the enzymatically active AChE was bound to lectins *Canavalia ensiformis* (Con A) and *Lens culinaris agglutinin* (LCA) that recognize terminal mannoses, whereas Western blot assays showed a very low percentage of AChE protein being recognized by the lectin. This indicates that active and inactive forms of AChE vary in their glycosylation pattern, particularly in the presence of terminal mannoses in active ones. Moreover, sAD subjects showed reduced binding to terminal mannoses compared to non-demented controls, while, for fAD patients that carry mutations in the PSEN1 gene, the binding was higher. The role of presenilin-1 (PS1) in modulating AChE glycosylation was then studied in a cellular model that overexpresses PS1 (CHO-PS1). In CHO-PS1 cells, binding to LCA indicates that AChE displays more terminal mannoses in oligosaccharides with a fucosylated core. Immunocytochemical assays also demonstrated increased presence of AChE in the trans-Golgi. Moreover, AChE enzymatic activity was higher in plasmatic membrane of CHO-PS1 cells. Thus, our results indicate that PS1 modulates trafficking and maturation of AChE in Golgi regions favoring the presence of active forms in the membrane.

## 1. Introduction

Alzheimer’s disease (AD), the most common cause of dementia among elderly people, is characterized by a compromise of cholinergic activity as a result of a decrease in the levels of the neurotransmitter acetylcholine and a reduction in acetylcholine synthetizing enzyme choline acetyltransferase, as well as hydrolyzing enzyme acetylcholinesterase (AChE) [1]. Intriguingly, despite this reduction in AChE enzymatic activity, estimation of the protein levels in the brain of AD patients demonstrated that total AChE protein levels are preserved [2]. Maintenance of high levels of AChE protein may account for preservation or increments in the pool of enzymatically inactive AChE species in the pathological brain. This imbalance between active/inactive AChE species has not been studied until now and may result from alterations in post-translational modifications of AChE that could alter the protein structure and conformation, affecting enzymatic function. In this regard, glycosylation is a common post-translational modification that affects protein structure and function [3]. AChE is a glycoprotein with three potential N-glycosylation sites that play an important role in proper folding, trafficking of the protein from endoplasmic reticulum (ER) to Golgi, as well as to achieve full enzymatic activity. Indeed, correct N-glycosylation is essential for AChE maturation and membrane targeting [4] since abnormally glycosylated forms are retained in the ER.

Alterations in AChE glycosylation have been described in some pathological conditions, such as Creutzfeldt–Jakob disease [5], breast cancer [6], and also in AD. Indeed, in the brain and cerebrospinal fluid (CSF) of AD patients, glycosylation of AChE enzymatic activity is altered [7] and expression of a particular AChE monomeric form with different glycosylation is increased [8,9]. Moreover, it has been suggested that the increase in AChE activity that occurs around amyloid plaques is due to AChE displaying differences in glycosylation, as compared with those from normal brain, and that these altered glycoforms may facilitate formation of amyloid fibrils in the AD brain [10].

Alterations in glycosylation in AD have also been reported for other AD key proteins, such as β-amyloid precursor protein (APP), tau, and β-secretase 1 (BACE1), and γ-secretase subunit nicastrin [11,12,13,14]. Altered glycosylation would affect multiple biological processes, such as neuroinflammation, cell adhesion, and cell signaling in AD brains [12]. In an altered glycosylation pattern in the brain of AD patients, presenilin 1 (PS1) could develop an important role. PS1 is the catalytic component of the γ-secretase complex participating in both amyloidogenic and non-amyloidogenic pathways of APP processing; thus, β-amyloid (or Aβ), the AD triggering effector [15,16], is a proteolytic peptide generated by processing of APP by successive action of two enzymes, β-secretase and γ-secretase (for a review, see [17]). Since mutations in PS1 are implicated in the development of most of the familiar forms of AD (fAD), malfunction of PS1 is associated with AD pathology [18]. On the other hand, it has also been suggested that PS1 can modulate glycosylation of diverse proteins, either by directly affecting the glycosylation process [12] or by regulating the cellular location of these glycoproteins [19,20]. In fact, PS1 has been implicated in glycosylation and maturation of nicastrin, another component of the γ-secretase complex [21], APP [19,20,22], and tropomyosin receptor kinase B (TrkB) [20]. Furthermore, it has been shown that PS1 modulates the intracellular trafficking and surface expression of a microglia receptor, the triggering receptor expressed on myeloid cells 2 (TREM2), also affecting its phagocytic function [23].

Previous reports from our group have demonstrated that PS1 can interact with AChE, showing an intracellular overlapping distribution in reticulum and Golgi [24]. PS1 influences the processing of the AChE membrane anchor PRiMA subunit [25], but it can also influence glycosylation of AChE [24]. In this regard, expression of the fAD-linked A246 human PS1 mutation in a transgenic mouse model leads to a reduction in AChE activity and alterations in AChE glycosylation, suggesting that loss of function of PS1 caused by fAD mutation may disturb AChE maturation [24]

In this study, we have investigated in brain cortex of AD, including fAD patients, alterations in glycosylation of both enzymatic active and total AChE protein by lectin binding assays. We have also analyzed the role of PS1 in the glycosylation, trafficking, and location of AChE in the plasmatic membrane in cellular models that stably overexpress PS1.

## 2. Results

### 2.1. AChE Enzymatic Activity and Protein Levels in AD Brain Cortices

AChE enzymatic activity, molecular forms pattern, and protein levels were analyzed in frontal cortex extracts of sAD and NDC subjects. In agreement with previous reports [2,9], AChE activity was decreased (~33% reduction, *p* = 0.041) in brain extracts from sAD subjects compared with NDC subjects (Figure 1A). Sucrose density gradients were employed to separate molecular forms of AChE and sedimentation profiles from sAD and NDC brain extracts (Figure 1B) showed that tetrameric (G4) species stand for the major peak of AChE activity with a minor contribution of lighter species (G1 + G2). As previously reported, a reduction in G4 was observed in gradients of AD [9]. A ratio G4/(G1 + G2), which represents the amount of tetrameric versus lighter forms, was calculated, resulting in a decrease in samples of AD patients (~17% decrease, *p* = 0.029). Brain cortex samples from patients of fAD were also analyzed, showing a higher reduction in AChE activity than sAD (~56% reduction with respect to NDC, *p* < 0.0001; ~34% reduction with respect to sAD, *p* = 0.006 Figure 1A) and a decrease in G4 forms, which, in turn, results in a decrease in the G4/(G1 + G2) ratio with respect to NDC (~40%, *p* < 0.001; Figure 1B).

AChE protein levels were analyzed by Western blotting using a specific anti-AChE antibody (A11) raised against a peptide that maps the exclusive C-terminus of the human cholinergic AChE-T variant, the main AChE specie in human brain [26]. This antibody resolved three major immunoreactive bands of approximately 75, 65, and 55 kDa (Figure 1C), consistent with previous observations [26]. There were no significant differences in the immunoreactivity levels of these AChE bands between NDC and AD or for fAD samples, which displayed a major decrease in AChE activity. These unaltered AChE protein levels in sAD and fAD did not parallel the decrease in AChE activity, indicating that most of the AChE in the human brain corresponds to inactive species or species with low activity.

### 2.2. Altered AChE Glycosylation in AD Brain Cortices

Given that glycosylation could determine maturation of AChE and achievement of enzymatic activity, we assayed lectin binding pattern of AChE in frontal cortex extracts from NDC, AD, and fAD patients. Lectin binding analysis determines the ability of AChE to bind to several immobilized lectins that recognize and bind to specific terminal sugar moieties of glycoproteins. The lectins used were Con A, which recognizes α-mannose residues; LCA, which also interacts with α-mannosyl residues of N-linked sugar chains but requires the presence of a core fucose residue α-linked; WGA, which recognizes N-acetyl-D-glucosamine and sialic acid; and SNA, which joins preferentially to sialic acid attached to terminal galactose.

First, a potential alteration in the glycosylation pattern of the enzymatically active AChE was determined by measuring enzymatic activity in the unbound fraction to the lectin. The results showed that, in control and pathological samples, most of the AChE activity was recognized by Con A and LCA, where the strongest interaction was observed with Con A (Figure 2A,B). With regard to binding to lectins that recognized sialic acid, a large proportion of activity was bound to WGA, whereas only a small percentage was bound to SNA (Appendix A). These results indicate that there is a higher proportion of active AChE displaying high-content mannoses residues on the surface. Furthermore, AChE activity in the unbound fraction was increased significantly (*p* < 0.05) in sAD samples with respect to NDC for Con A and LCA lectins (Figure 2A,B), indicating the presence of alterations in the glycosylation pattern. However, in samples from fAD patients, no significant changes were observed, as compared with NDC, in binding to lectin Con A, although a significant decrease was observed in the fraction not recognized by LCA (~30% reduction; *p* = 0.028; Figure 2A,B). Interestingly, the AChE activity levels unbound to both Con A and LCA were significantly lower in fAD samples than in sAD (*p* = 0.001 for Con A and *p* = 0.03 for LCA), indicating differences in mannose glycosylation between fAD and AD cortices. These divergences may account for the dissimilarities in the molecular form distribution between sporadic and familiar cases.

In this regard, since AChE displays a complex molecular form pattern in the brain, we additionally analyzed the interaction of each molecular form with lectins Con A and LCA. Sedimentation profiles of the fractions unbound to the lectins showed that both G4 and lighter G1 + G2 forms were recognized by Con A, with no apparent differences between NDC and pathological samples (Figure 2C). In the case of LCA, profiles of the unbound fraction (Figure 2D) showed that most of the G4 was recognized by the lectin; however, a percentage of the lighter G1 + G2 remained in the fraction unbound to the lectin, which tends to be particularly abundant in sAD (*p* = 0.1 with respect to NDC).

Next, in an attempt to address the potential alteration in glycosylation of the inactive pool, the binding pattern of AChE protein to the different lectins was analyzed by Western blotting. Immunoreactive bands of 75, 65, and 55-kDa were resolved in the unbound fractions to the lectins, whereas only the 75 kDa species was resolved in the fractions recognized by the lectin (Figure 2E,F). Thus, we performed a quantitative analysis of the percentages of the 75-kDa immunoreactive band recognized by the lectins. Interestingly, while enzymatic activity assays showed that most of the AChE activity was bound to Con A and LCA, most of the AChE protein estimated by Western blot remained in the fractions unbound to lectins (Figure 2E,F). This disparity between the percentages of activity and AChE protein recognized by these lectins would indicate that active forms of AChE differ from inactive forms in their glycosylation, particularly with the exclusive presence in the active forms of oligosaccharides with high content of terminal mannose residues.

Comparisons of binding to lectins between NDC and sAD groups were performed and differences in the glycosylation of the 75-kDa band were noticed. In sAD samples, the percentage of 75 kDa immunoreactive band that binds to Con A (~70% decrease, *p* = 0.025; Figure 2E) and LCA (~45%, *p* = 0.041; Figure 2F) was lower than in NDC. However, in fAD, a different trend was observed, a significantly more AChE protein was recognized by LCA as compared with NDC (~80% increase, *p* = 0.031; Figure 2F), and significant increases with respect to sAD for both LCA (~64% *p* = 0.00; Figure 2F) and Con A binding (~50%, *p* = 0.01; Figure 2E) were noticed.

We also analyzed binding of AChE immunoreactive species to SNA and WGA. The results revealed that the major proportion of AChE protein remained in the unbound fraction to WGA and SNA in both NDC and sAD samples (Appendix A). Binding to WGA showed that, whereas a high proportion of AChE activity was recognized by the lectin, only a small proportion of the protein was detected in the bound fraction (Appendix A). A comparison of the binding of AChE to the two lectins between NDC and sAD samples showed no alterations in pathological samples.

### 2.3. PS1 Influences AChE Glycosylation in a CHO-PS1 Cellular Model Overexpressing the Cholinergic Species

Our glycosylation analysis indicated that AChE displays particular glycosylation in sAD and fAD patients in which the fAD cases are caused by mutations in PSEN1 gene. Previous reports point toward PS1 acting as a chaperone involved in regulation of trafficking and maturation of glycoproteins [20]. Therefore, we used cellular models to investigate whether PS1 could modulate AChE glycosylation.

Since Chinese hamster ovary (CHO) cells display similar glycosylation pathways to human cells [27], glycosylation of AChE was analyzed in CHO cell lines. We compared AChE glycosylation between a CHO cell line with a wild-type phenotype (CHO-wt) and a cell line that stably overexpresses human wild-type PS1 and wild-type AβPP (CHO-PS1; [28] (see Appendix A for the assessment of PS1 overexpression)). Given that CHO cells express low levels of AChE [25], both lineages were transfected with a cDNA coding for human AChE-T, the major variant in the human brain, together with a cDNA that encodes PRiMA1, the anchoring subunit of AChE-T to the plasmatic membrane. The efficacy of the transfection was confirmed measuring AChE and PRiMA1 expression in cellular extracts. There was a significant increment in AChE activity along with increased levels of a 70 kDa AChE immunoreactive band in extracts of CHO cells transfected with AChE-T+PRiMA, as compared with PCI-control and also compared with cells transfected only with AChE-T (Appendix A). PRiMA1 is the limiting factor for the presence of cholinergic tetrameric AChE forms in the plasma membrane [29], and sedimentation analysis indicated that only cells co-transfected with AChE-T+PRiMA1 displayed substantial levels of tetrameric AChE (Appendix A), the major AChE specie expressed in human brain extracts (see Figure 1B). The efficiency of AChE and PRiMA overexpression and tetrameric generation was similar in both cell lines CHO-wt and CHO-PS1 (Appendix A).

Therefore, lectin-binding assays were performed in extracts from CHO-wt and CHO-PS1 cells transfected with AChE-T+PRiMA (Figure 3). After interaction with lectins Con A, LCA, and SNA, AChE activity was assayed in fractions not recognized by the lectins. According to the results obtained in brain extracts, most of the AChE activity was recognized by Con A, displaying only a residual amount of activity in the unbound fraction, with no differences between CHO-wt and CHO-PS1 cells (Figure 3A). The sedimentation profile of the Con A unbound fraction in both cell lines showed that G4 forms were fully recognized by the lectin and only a small portion of lighter G1 + G2 forms remained unbound (Figure 3B). LCA also displayed strong binding, although the activity in the unbound fraction was significantly lower in CHO-PS1 (~38% reduction, *p* = 0.02) than in wild-type cells (Figure 3C). Likewise, most of the G4 forms bound to LCA, while the unbound AChE corresponded to lighter G1 + G2 forms (Figure 3D). Regarding binding to SNA, most of the AChE activity remained in the unbound fraction to the lectins (CHO-wt, 96.5 ± 2.1, %; CHO-PS1, 97.1 ± 7.5, %). Thus, oligosaccharides of active AChE displayed a fucosylated core and are rich in terminal mannose residues, and even more so in CHO-PS1 cells, with no sizeable occurrence of terminal sialic acid residues.

We then decided to assess binding of AChE total protein to lectins LCA and Con A by Western blotting. The interaction of AChE with Con A exhibited a significant reduction in the 70-kDa immunoreactive band on the unbound fraction of CHO-PS1 (~63% reduction, *p* = 0.003) compared to CHO-wt cells (Figure 3E). CHO-PS1 also displayed a reduction in 70-kDa AChE immunoreactive species that remained in the unbound fraction to LCA (~54% reduction; *p* = 0.001; Figure 3F) with respect to CHO-wt. Therefore, we conclude that AChE in cells that overexpress PS1 presents more N-glycans with a fucose core and displays terminal mannoses. Since these sugar residues are added to glycans in distal regions of the Golgi network, PS1 could promote the presence and glycosylation of AChE in distal regions of Golgi.

CHO-PS1 cells overexpress human wild-type APP together with PS1, leading to an increase in generation of Aβ42 peptide. It has been demonstrated that Aβ42 treatment alters AChE expression and glycosylation [8]. Therefore, we analyzed AChE glycosylation in CHO-wt cells that were treated with 5 μM Aβ42 for two consecutive days. LCA binding showed no alterations in either AChE activity or protein levels in the unbound fraction with the Aβ42 treatment (Appendix A).

On the other hand, LCA interaction was assayed in CHO-PS1, in which PS1 overexpression was knocked down using PS1 siRNA. The PS1 levels were measured and the results demonstrated a steep reduction in PS1 expression 48 h after siRNA transfection (Appendix A). LCA binding assays showed a trend to increase in the amount of AChE enzymatic activity and protein not recognized by the lectin in cells with reduced PS1 levels (Supplemental Appendix A).

### 2.4. PS1 Drives AChE in the Trans-Golgi Network in CHO-PS1 Cells and Favors the Presence of Active AChE Species at the Plasmatic Membrane

Since our results suggest that PS1 promotes glycosylation of AChE in the trans-Golgi network (TGN) regions, immunocytochemical assays were performed in CHO-wt and CHO-PS1 cells that overexpress AChE-T+PRiMA to analyze the location of AChE in these TGN regions. Confocal assays showed that AChE colocalized with trans-Golgi marker (TGN-46) in both CHO-wt and CHO-PS1 cells (Figure 4). The calculated Mander’s co-localization coefficient was significantly higher (*p* = 0.003) in CHO-PS1 cells than in CHO-wt cells, indicating that the amount of AChE present in TGN was greater in PS1 overexpressing cells.

The AChE increase in the TGN could represent more mature enzymatic active cholinergic species, which will be finally located at the plasmatic membrane. We analyzed whether PS1 could influence the presence of AChE in the plasmatic membrane. For this, AChE enzymatic activity in the membrane was determined by an adapted Ellman method on the surface of CHO-wt and CHO-PS1 cells and appeared to be higher in cells that overexpress PS1 (~66% increment; *p* < 0.001; Figure 5A). Furthermore, biotinylation of cell surface proteins was performed followed by subsequent Western blots showing that the percentage of the AChE protein bound to biotin was unaltered in CHO-PS1 with respect to CHO-wt cells (Figure 5B), indicating that only the levels of active AChE located at the plasma membrane were increased. Moreover, confocal assays were carried out to analyze the location of AChE in the plasmatic membrane, and similar Mander’s coefficients were obtained in CHO-wt and CHO-PS1 for co-localization of AChE and membrane marker ATPase (Figure 5C), indicating that the amount of AChE protein in the plasmatic membrane is not altered by PS1. In addition, targeting of fucosylated AChE in the membrane was achieved by immunocytochemistry and confocal assays using a conjugated-LCA lectin along with anti-ATPase and AChE antibodies. Fucosylated AChE was defined by co-localized fluorescence for LCA and AChE, and the location in membrane was then analyzed for co-localization with ATPase. Mander’s coefficient for CHO-PS1 was significantly higher compared to CHO-wt cells (*p <* 0.0001; Figure 5D), indicating that the amount of AChE with core fucosylated mannose residues is higher in CHO-PS1 cells.

In conclusion, our results reveal that PS1 favors the maturation and core fucosylation of AChE in the distal Golgi and its location to the plasma membrane in CHO cells, thus modulating the localization of AChE active forms.

## 3. Discussion

In this study, we have seen that active and inactive forms of AChE differ in their glycosylation pattern, with the presence of oligosaccharides that contain terminal mannoses in active species. Interestingly, this mannose glycosylation pattern associated with acquisition of enzymatic activity appears to be altered in brain cortex of sAD patients compared with controls but also appears to differ with regard to fAD. The occurrence of fAD mutations in PS1 gene, together with previous data [24], strongly suggests a role for PS1 in AChE glycosylation. Hence, we have demonstrated alterations in AChE glycosylation in CHO cells that overexpress PS1, which, in turn, indicate that PS1 modulates the trafficking in Golgi regions and location of active AChE variants in the plasmatic membrane.

In the brain cortex of AD patients, our results corroborate the reduction in enzymatically active AChE tetrameric forms, whereas AChE protein levels that were mostly attributed to an inactive pool were preserved [2]. The inactive AChE forms in the brain have been previously identified as AChE-T splicing variants [26]. The splicing of the ACHE gene generates different transcripts with distinct N- and C-terminal peptides that determine the ability of the molecule to form oligomers, yet they all share the same catalytic domain [30]. In the brain, AChE-T transcript is the most abundant and generates monomeric subunits that could be organized into dimers and tetramers. The tetramers anchored to the membrane are considered to be the cholinergic functioning forms [29,30,31,32]. Therefore, depletion of enzymatically active AChE species could determine the cholinergic impairment present in AD. Nonetheless, the biological significance of the inactive pool of AChE is unknown.

The inactive pool of AChE has been attributed to monomeric species that could act as reservoirs for further assembly of cholinergic tetramers, or other exportable species that could play non-cholinergic roles. However, the evidence indicates that most of the inactive AChE molecules are rapidly degraded and do not transit the Golgi apparatus [33]. Moreover, inactive AChE-T species may be in part due to inefficient post-translational processing of the AChE subunits within the AD brain, which, in turn, compromises the ability to form the active complex as acquisition of catalytic activity is significantly affected by glycosylation [4]. Thus, an imbalance of active/inactive species in the AD brain may be not only due to depletion of cholinergic neurons in the brain of affected subjects [34] but also to an altered glycosylation pattern of AChE.

AChE contains three N-glycosylation sites and mutants that express defects in one or all the sites showing reduced or absent AChE activity [4]. Indeed, a lack of glycosylation leads to misfolding and failure to export AChE to the plasma membrane, reducing enzymatic activity [35,36]. In this study, we have analyzed, via lectin binding assays, glycosylation of AChE enzymatic activity and protein. Glycosylation assays showed that active AChE displays terminal mannoses, and, given the high binding to LCA, the results indicate that most forms consist of mature oligosaccharides. Furthermore, a large proportion of the active AChE presented glycans contained terminal N-acetylglucosamine and/or sialic acid, as suggested by the binding to WGA. On the contrary, the interaction with SNA suggested that just a small percentage of active AChE species presented terminal sialic acid.

AChE is a polymorphic protein that can be assembled as different molecular forms of the same catalytic subunit. Moreover, diverse molecular forms may display distinct patterns of glycosylation in the same cell [29]. In our study, sucrose gradients showed that all active forms presented mannose-containing oligosaccharides, whilst mature glycans that contain mannoses with a fucose core are present in a larger proportion as tetrameric forms rather than monomers. It may be possible that the different glycosylation of the lighter forms could differentiate the molecular forms that are precursors of more complex forms. Hence, in the human brain, enzymatically active AChE displayed a complex glycosylation pattern that differed between molecular forms [9]. However, in the present study, lectin binding of AChE protein, which comprises active and inactive species, showed differences compared to the results obtained by determining enzymatic activity, with only a small percentage of the protein displaying oligosaccharides with terminal mannoses and sialic acid. Hence, non-active AChE is less recognized by the lectins used in this study than active AChE. We can speculate that, in pathological conditions, alterations in glycosylation processing of non-active AChE impairs correct folding of the protein and acquisition of proper enzymatic activity.

More relevantly, the glycosylation pattern of AChE in the brain resulted in altered sAD samples as the percentage of both enzymatically active AChE and the 75-kDa protein recognized by Con A and LCA were lower than in NDC. Thus, in the sAD cortex, there was a reduction in the active AChE glycoforms that displayed terminal mannoses. It is expected that glycosylation of mature, active AChE is completed in Golgi, and glycans recognized by LCA should be added in trans-Golgi regions, suggesting that the alterations in AChE glycosylation detected in sAD might occur at this subcellular level.

Interestingly, binding of AChE to the lectins was different in samples from fAD patients compared to sAD since the fraction of AChE with mannose residues in fucosylated oligosaccharides was higher in these fAD samples. Our results suggest a particular mechanism in fAD that would affect glycosylation of AChE. The fAD samples present mutations in the gene that codifies PS1, and, since it has been suggested that PS1 influences glycosylation of proteins, such as nicastrin [37] and neural cell adhesion molecule (NCAM) [38], we hypothesize that alterations in PS1 could affect AChE glycosylation. Indeed, previous studies of our group demonstrated that PS1 physically interacts with AChE and that this interaction, when resolved by co-immunoprecipitation, is lower in fAD than in sAD [24]. Moreover, transgenic mice that express the fAD PS1-A246E mutation showed an impairment in AChE maturation and glycosylation [24].

We further analyzed the influence of PS1 in the glycosylation of AChE-T variants using a cellular model (CHO-PS1) that stably overexpressed human wild-type PS1 and APP [28]. In these CHO cells, co-transfection of AChE-T and PRiMA1, subunits of cholinergic AChE tetramers, leads to synthesis of cholinergic G4-PRiMA-linked forms, as confirmed by sedimentation analysis on sucrose density gradients. This profile of molecular forms of AChE in CHO cells resembles the profile of human brain cortices (compare Figure 1 and Figure 3). In the CHO-PS1 cells, AChE glycosylation was different to wild-type, with a higher amount of AChE (active and total protein) exhibiting oligosaccharides that displayed mannoses with a fucosylated core. As previously mentioned, core fucose residues are added in trans-Golgi regions; thus, our results indicate that PS1 would favor glycosylation of AChE-T in these subcellular regions. Indeed, co-localization assays of AChE-T with a TGN marker confirmed that the amount of AChE in trans-Golgi was higher in cells that overexpress PS1. We, therefore, speculate that PS1 could act as a chaperone of AChE (and other glycoproteins) trafficking along these biosynthetic compartments. Previous reports situate PS1 and AChE co-localizing in COS-7 cells throughout the endoplasmic reticulum and Golgi apparatus [24]. In these compartments, PS1 can mediate N-glycosylation of diverse glycoproteins [12,39]. PS1 could affect the subcellular location of these glycoproteins, which will influence the N-glycosylation process. In this regard, a recent study has shown that PS1 overexpression increases the localization of BACE1 in the endoplasmic reticulum and Golgi [40], indicating again a role for PS1 in processing, maturation, and transportation of glycoproteins. We could not discount that PS1 may have a direct effect on N-glycosylation of a subset of glycoproteins, which will affect their final, and functional, subcellular location [12]. In the AD context, impaired PS1 function modulating glycosylation may be physio-pathologically relevant as a widely investigated role in APP proteolytic processing and deserve further research.

Moreover, an indirect influence of PS1 on AChE glycosylation through Aβ should be considered. Several studies have demonstrated that Aβ is able to alter glycosylation of proteins, such as reelin [41], and even AChE active monomers [9]. In CHO-PS1, as a result of the overexpression of PS1 and APP, increased Aβ generation is expected [42], but, in our present study, wild-type cells treated with Aβ42 did not display alterations in glycosylation. These results do not discount the potential influence of Aβ on AChE glycosylation but point toward a direct role of PS1 in AChE glycosylation, which is independent of β-amyloid. Moreover, the role of PS1 in AChE glycosylation was also indicated by a reduction in LCA binding of AChE when PS1 expression is partially diminished following treatment with PS1 siRNA.

The mature glycosylated cholinergic AChE Is finally transported out of the Golgi/TGN and located in the plasmatic membrane to properly carry out its physiological function. In our study, we reported that the AChE enzymatic activity at plasma membrane was increased in cells that overexpress PS1, although AChE protein levels did not change with respect to wild-type cells. Interestingly, confocal assays showed that the amount of AChE located at the membrane that contains mannose residues in fucosylated core glycans was higher in CHO-PS1 cells. Thus, PS1 could modulate the trafficking and final location of active forms of AChE in the plasmatic membrane. This active AChE contains core fucosylated oligosaccharides that present terminal mannose residues, indicating a role of PS1 in maturation of active AChE in TGN. Nonetheless, we could not discount that PS1 may affect AChE internalization from the cell surface or modulate recycling from endosomes and trans-Golgi to the cell surface. PS1 localized at the endoplasmic reticulum is not associated with γ-secretase activity [43,44], supporting a regulatory transport function at this step in the secretory pathway [45].

In conclusion, PS1 can regulate maturation and trafficking of AChE, modulating its glycosylation. Since we demonstrated that glycosylation differs between active and inactive forms of AChE, we inferred that PS1 could influence the proportion of active AChE. In fAD caused by mutations in PS1, defective regulation could have physio-pathological implications, resulting in altered intracellular transport of AChE, which has also been demonstrated for other key glycoproteins in AD, such as TREM2 [23] and BACE1 [40]. In our study, AChE from fAD patients showed a glycosylation pattern different to sAD that could be the result of alterations in trafficking mediated by PS1. In sAD, a multifactorial pathology, the changes in AChE glycosylation are more complex to decipher and could also reflect alterations in molecular form patterns, with a specific decrease in tetrameric forms. Further studies are needed to decipher the role of PS1 as a potential chaperone of the cellular location of other glycoproteins. Altered glycosylation of a key subset of glycoproteins may also influence APP proteolytic processing and Aβ generation, having a role in AD pathogenesis.

## 4. Materials and Methods

### 4.1. Human Brain Samples

This study was approved by the ethic committee of the Hospital General Universitario de Elche and performed according to the Declaration of Helsinki. A collection of frozen frontal cortex (Brodmann areas 9/10) samples from sporadic AD (sAD) patients [n = 4, 2 male/2 female; 88 ± 6 years], non-demented control (NDC) cases [n = 4, 2 male/2 female; 60 ± 6 years] was obtained from the UIPA Neurological Tissue Bank (Unidad de Investigación Proyecto Alzheimer, Madrid, Spain). After neuropathological examination, AD cases were categorized as stages V–VI of Braak (Braak and Braak, 1998). NDC samples correspond to individuals with no clinical dementia and no evidence of brain pathology. Frontal cortex samples of four fAD patients, carriers of PSEN1 mutations as indicated: two PS1-V89L cases [male, 54 years old (y), Braak stage V; mal, 57 y Braak stage VI], one PS1-M139T case (male, 64 y, Braak stage V), and one PS1-E318G case (male, 54 y, Braak stage IV] were also included in the study. These fAD samples were obtained from the Banc of Teixits Neurològics, Universitat de Barcelona-Hospital Clínic (Barcelona, Spain). The mean post-mortem interval of all tissues was between 1.5 and 6 h, with no significant differences between groups.

Approximately 0.1 g of human frontal cortex samples (stored at −80 °C) were thawed slowly at 4 °C and homogenized (10% *w*/*v*) in 50 mM Tris-HCl (Cat. No. T1503: Sigma-Aldrich Co., St. Louis, MO, USA), pH 7.5, 500 mM NaCl (Cat. No. S9888: Sigma-Aldrich Co.), 5 mM EDTA (Cat. No. ED-500: Sigma-Aldrich Co.), 1% (*w*/*v*) Nonidet P-40 (Cat. No I3021: Sigma-Aldrich Co.), 0.5% (*w*/*v*) Triton X-100 (Cat. No. T9284: Sigma-Aldrich Co.) and complemented with a protease inhibitor cocktail [9]. Homogenates were sonicated and centrifuged at 70,000× *g* at 4 °C for 1 h, and supernatants collected and frozen at −80 °C until assayed.

### 4.2. Cell Culture

Chinese Hamster Ovary cells with wild-type phenotype (CHO-wt) were grown in D-MEM+GlutaMAX™-I (Dulbecco’s Modified Eagle Medium; Gibco^®^, Life technologies Paisley, UK) supplemented with 10% fetal bovine serum (FBS, Gibco^®^) and 1% penicillin/streptomycin solution (P/S; 100 U/mL, 100 µg/mL, Gibco^®^). CHO cells stably overexpressing wild-type human PS1 and APP [CHO-PS1, a generous gift from Dr. Selkoe; see [28] were grown in D-MEM+GlutaMAX™-I supplemented with 10% FBS, 1% P/S, G-418 (200 µg/mL), and puromycin (2.5 µg/mL).

CHO-wt and CHO-PS1 cells were seeded at a density of 5 × 105 cells on 35 mm tissue culture dishes and transfected the following day with plasmid cDNA using Lipofectamine™ 3000 (Invitrogen™, Life Technologies, Paisley, UK) according to the manufacturer’s instructions. The plasmids employed encoded: PCI “empty” vector (Promega, Madison, WI, USA) serving as a negative control; human AChE-T variant, the most common in the brain, under the cytomegalovirus (CMV) promoter-enhancer (a generous gift from Dr. H. Soreq, The Institute of Life Sciences, The Hebrew University of Jerusalem, Jerusalem, Israel); and PRiMA-HA complementary DNA containing the full-length mouse PRiMA isoform I with a hemagglutinin (HA) epitope inserted before the C-terminus stop codon (a generous gift from Dr. K. Tsim, Department of Biology and Center for Chinese Medicine, The Hong Kong University of Science and Technology Hong Kong, China). Cells were transfected with 4 µg of PCI or 4 µg of AChE-T or 2 µg of AChE-T plus 2 µg PRiMA-HA.

To knock down the PS1 gene expression in CHO-PS1, cells were transfected with 50 nM human PS1 siRNA (Cat. No sc-36312, Santa Cruz, Dallas, TX, USA or BLOCK-iT™ Alexa Fluor™ Red Fluorescent Control Oligo 50 nM (Cat. No 14750100, Invitrogen, Waltham, MA, USA) as a negative control. The transfection was repeated the following day at a lower concentration of 30 nM. After 24 h, the cells were solubilized. Uptake of BLOCK-iT by cells was assessed using fluorescence microscopy.

For treatment with Aβ synthetic peptide, Aβ1–42 (Aβ42) (American Peptide Co., Inc., Sunnyvale, CA, USA) and the scrambled control peptide (AIAEGDSHVLKEGAYMEIFDVQGHVFGGKIFRVVDLGSHNVA) were dissolved in sterilized distilled water at a concentration of 1 mg/mL, aliquoted, and stored at −80 °C. Suspensions of Aβ42 or the scrambled peptide corresponding to a final concentration of 5 μM were added to each culture 4 h after the AChE and PRiMA transfection and the treatment was repeated 24 h later. Once Aβ42 preparations had been added to the cells, the medium was no longer changed.

After 48h of AChE plus PRiMA transfection, cells were washed with phosphate–saline buffer (PBS) and resuspended in 120 µL ice-cold extraction buffer: 50 mM Tris-HCl, pH 7.5, 150 mM NaCl, 5m M EDTA, 1% (*w*/*v*) Nonidet P-40, 0.5% (*w*/*v*) Triton X-100 and supplemented with the protease inhibition cocktail as mentioned above. Cell lysates were sonicated and centrifuged at 70,000× *g* at 4 °C for 1 h, and then supernatants were collected and frozen at −80 °C until assayed. AChE activity was measured to determine transcription efficacy.

### 4.3. AChE Assay and Total Protein Determination

AChE activity was determined by a modified microassay version of the colorimetric Ellman method [46]. AChE was assayed with 1 mM acetylthiocholine iodide (Cat. No A-5751, Sigma-Aldrich Co.) in the presence of 50 µM tetraisopropyl pyrophosphoramide (Iso-OMPA, Cat. No T1505, Sigma-Aldrich Co.) to block any contamination from butyrylcholinesterase. One milliunit (mU) of AChE activity was defined as the number of nmoles of acetylthiocholine hydrolyzed per min at 22 °C.

Enzymatic activity from the membrane surface of the cells was measured in wells with attached intact cells. For that, wells were washed with PBS and then Ellman’s assay was performed. Then, cells were washed to eliminate any residue left from the enzymatic activity assay and protein concentration was determined in the well to normalize the results obtained.

Total protein concentrations were determined using the bicinchoninic acid method, with bovine serum albumin (BSA) as standard (Cat. No 23225, ThermoFisher, Waltham, MA, USA), according to manufacturer’s instructions.

### 4.4. Sedimentation Analysis

Molecular forms of AChE in brain and cellular extracts were separated according to their sedimentation coefficients by ultracentrifugation on 5–20% (*w*/*v*) continuous sucrose gradients in 50 mM Tris-HCl (Ph 7.4) containing 150 mM NaCl, 50 mM MgCl2 (Cat. No M9272, Sigma-Aldrich Co.), and 0.5% Triton X-100. Equal sample volumes were carefully added onto the top of the gradient and ultracentrifugation was carried out for 4 h at 250,000× *g* and 4 °C in a Beckman TLS 55 rotor. After centrifugation, 30 fractions were gently collected from the top of the tube and assayed for AChE activity to identify individual AChE forms (G4 = tetrameric species, G1 + G2 = monomers plus dimers). Catalase (11.4S) and alkaline phosphatase (6.1S) were employed as markers of known sedimentation coefficient to identify AChE molecular forms.

### 4.5. Cell-Surface Biotinylation Assay

The EZ-Link™ Sulfo-NHS-SS Biotinylation Kit (Cat. No 21445, ThermoFisher) was used to determine the presence AChE on cell surface in CHO-wt and CHO-PS1 cells transfected with AChE plus PRiMA cDNAs. Briefly, cells were incubated with EZ-Link™ Sulfo-NHS-SS-Biotin (Cat. No 21331, ThermoFisher) for 30 min at 4 °C. Then, biotinylated cells were quenched with PBS plus 50 mM NH4Cl (Cat. No A9434, Sigma-Aldrich). Afterward, cells were solubilized with the cell lysis buffer and centrifuged at 10,000× *g* for 2 min at 4 °C to obtain the supernatant, which was incubated with the NeutrAvidin overnight at 4 °C. Unbound fraction to NeutrAvidin was obtained after centrifugation 30 s at 6000× *g* and the supernatant stored. Proteins bound with NeutrAvidin were eluted with Laemmli-sodium dodecyl sulphate (SDS) buffer (Cat. No 15403949, ThermoFisher) plus 50 mM dithiothreitol (DTT, Cat. No DTT-RO, Roche, Basel, Switzerland). The bound and unbound AChE species were assayed by Western blotting.

### 4.6. Lectin Binding Analysis of AChE

Analysis of AChE glycosylation was performed by lectin binding assays. The Sepharose-conjugated lectins employed were: *Canavalia ensiformis* (Con A, Cat. No C9017, Sigma-Aldrich Co.), *Lens culinaris agglutinin* (LCA, Cat. No L0511, Sigma-Aldrich Co.), *Triticum vulgaris agglutinin* (wheat germ, WGA, Cat. No L1394, Sigma-Aldrich Co.), and *Sambucus nigra* (SNA, Cat. No AL-1303-2, Vector laboratories, Newark, CA, USA). Aliquots of 300 μL of brain extracts or 120 μL of cell homogenates were mixed with 80 μL or 50 μL of immobilized lectins (equal amount of protein to each lectin), respectively, and incubated overnight at 4 °C with gentle agitation. AChE-lectin complexes were separated from free AChE by centrifugation, and bound AChE was eluted from the lectin by incubation with Laemmli SDS sample buffer for 5 min at 98 °C. The AChE activity unbound to the lectin was analyzed in the supernatant fraction and both the bound and unbound AChE species were assayed by Western blotting.

### 4.7. Western Blotting Assays

AChE, PRiMA-HA, and PS1 were analyzed by immunoblotting after sodium dodecyl sulphate-polyacrylamide electrophoresis (SDS-PAGE) under fully reducing conditions. Prior to electrophoresis, samples were denatured by heating at 98 °C for 7 min to specifically target AChE and at 50 °C for 15 min to PS1. Samples of brain or cell extracts were resolved on 8.5% or 10% SDS-polyacrylamide slab gels, respectively, loading 50 μg of protein. For lectin binding and biotinylated assays, equal amounts of the extract and unbound fraction and equivalent volume of the bound fraction were resolved.

Following electrophoresis, proteins were blotted onto nitrocellulose membranes (0.2 µm, Cat. No 1620168, Bio-Rad, Hercules, CA, USA), blocked with commercial blocking buffer (Cat. No 927-70001, LI-COR Biosciences, Lincoln, NE, USA), and probed with the following primary antibodies: anti-C-terminal AChE antibody (A11, RRID: AB_10917070, Santa Cruz); anti-HA antibody (HA, RRID: AB_260070, Sigma-Aldrich Co.); anti-PS1 antibody (against loop a.a. 275-367, RRID: AB_91785, Sigma-Aldrich Co.). A rabbit anti-glyceraldehyde 3-phosphate dehydrogenase antibody (GAPDH, RRID: AB_307275, Abcam, Cambridge, UK) was used as loading control. Western blots for different antibodies were performed separately to avoid re-probing of blots. Antibody binding was detected with the corresponding conjugated secondary antibody (IRDye 800CW goat anti-mouse IgG and IRDye 680CW goat anti-rabbit; RRID: AB_2687825 and AB_10956166, respectively, LI-COR Biosciences) and visualized on an Odyssey CLx Infrared Imaging System (RRID: SCR_014579, LI-COR Biosciences). Densitometric quantification of the signal from immunoreactivity bands was obtained employing LI-COR Software (Image Studio, RRID: SCR_015795, LI-COR Biosciences). For semi-quantitative analysis of AChE protein in brain samples, protein levels were normalized to GADPH.

### 4.8. Confocal Microscopy

Cellular localization of AChE was analyzed by immunocytochemistry in CHO-wt and CHO-PS1 cells. Cells were seeded at a concentration of 50,000 cells/well in a 12-well sterile culture plate containing one 18 mm diameter glass coverslip per well followed by transfection with both AChE and PRiMA cDNA. After 48 h, cells were fixed with methanol (Cat. No 8402.2500, J.T. Baker, Radnor, PA, USA) and after blocked with 2% BSA (Cat. No A3059, Sigma-Aldrich Co.) PBS-based buffer with 3.25 mM digitonin (Cat. No D141, Sigma-Aldrich Co.). Then, they were incubated with the primary antibodies: an anti-Na+/K+ ATPase antibody (ATPase, RRID: AB_1310695, Abcam); Anti-C-terminal AChE antibody (A11, RRID: AB_10917070, Santa Cruz); or anti-trans-Golgi network protein 2 (TGN46, RRID: AB_10597396, Proteintech, Rosemont, IL, USA). Then, incubation with the corresponding secondary antibody (Alexa Fluor^®^ 488 goat anti-rabbit IgG, Cyanine3 goat anti-mouse, Cyanine5 goat anti-mouse; RRID: AB_143165, AB_2534030, AB_2534033, respectively; Invitrogen) was completed. For glycosylation imaging, fixed cells were incubated with DyLight^®^ 649-LCA conjugated (Cat. No DL-1048-1, Vector Laboratories) for 15 min at room temperature prior to blocking.

To analyze location of proteins in plasmatic membrane, the blocking buffer did not include digitonin to avoid permeation of the membrane. Dual and triple immunofluorescence images were captured with sequential scans using a Leica laser-scanning spectral vertical confocal microscope (Leica TCS SP2, RRID: SCR_020231, Leica) and images digitalized and adjusted using Imaris (v9.3, RRID: SCR_007370) software.

### 4.9. Statistical Analysis

All data were analyzed with GraphPad Prism 7.0 (GraphPad Software Inc., San Diego, CA, USA). The Shapiro–Wilk test was used to analyze the distribution of each variable. The statistical analyses performed for parametric data were the Student’s unpaired *t*-test for comparisons between two groups, or by one-way ANOVA followed by a Tukey multiple comparisons test when three groups were compared. Non-parametric data were compared using the Mann–Whitney test for comparisons between two groups, and the Kruskal–Wallis test followed by a Dunn’s multiple comparisons test when three groups were compared. Results were presented as mean ± standard error of the mean (SEM), and *p* values < 0.05 were considered significant. Atypical data were removed from analysis using the ROUT method (Robust regression and Outlier removal), Q = 1%.

## Figures and Tables

**Figure 1 ijms-24-01437-f001:**
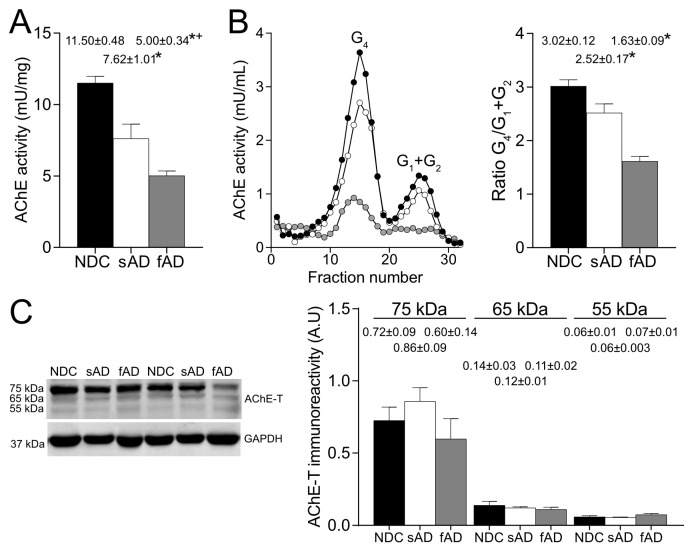
Brain AChE enzymatic activity, molecular forms pattern, and protein levels. (**A**) AChE enzymatic activity expressed in mU/mg of total protein measured in brain extracts from non-demented controls (NDC), sAD, and fAD patients (n = 4 for each group). (**B**) Molecular forms of AChE were separated by ultracentrifugation on sucrose gradients and were identified (tetramers: G4; light monomers and dimers: G1 + G2) in each fraction by comparison with the position of molecular weight markers catalase (11.4S) and alkaline phosphatase (6.1S). Representative profiles for NDC (●), AD (○), and fAD (●) patients are shown. A (G4)/(G1 + G2) ratio was calculated (right panel). (**C**) Representative Western blot showing immunoreactive bands for AChE at 75, 65, and 55 kDa and the loading control protein, GAPDH, at 37 kDa. The antibody used for AChE determination (A11) was specific for the AChE-T transcript. To the right, densitometric quantification of AChE-T immunoreactive bands in brain extracts normalized to the housekeeping protein GAPDH. The graphs represent mean ± SEM. Significantly different (one-way ANOVA test with Tukey’s multiple comparisons, *p <* 0.05) from the NDC group (*) or from sAD (+) is indicated.

**Figure 2 ijms-24-01437-f002:**
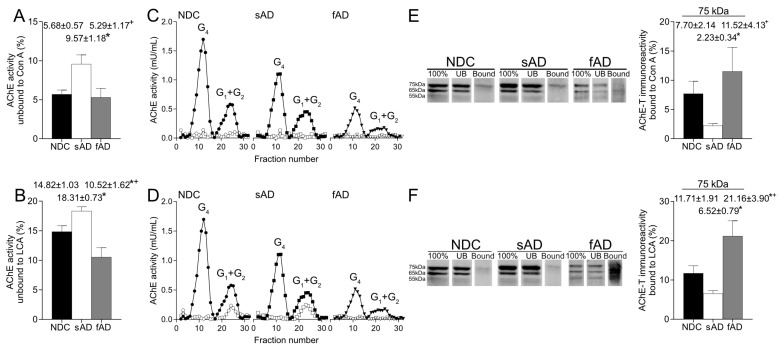
Glycosylation of active and inactive brain AChE species. Lectin-binding analyses of brain cortices extracts from non-demented controls (NDC), sAD, and fAD (n = 4 for each group) were performed by interaction with *Concanavalina ensiformis* (Con A) and *Lens culinaris* (LCA) lectins. After incubation of brain extracts with the lectins, the unbound fraction (the fraction not recognized by the lectin) was separated from the bound fraction by centrifugation. AChE enzymatic activity was measured in the fraction not recognized by (**A**) Con A or (**B**) LCA. Results are expressed as percentage over the total. The binding of AChE molecular forms with the lectins (**C**) Con A or (**D**) LCA was analyzed by comparison of the sedimentation profiles of brain extracts prior to interaction (closed symbols) with the profiles of non-recognized unbound fractions (open symbols). Representative profiles of molecular forms of AChE in NDC, sAD, and fAD are shown. Interaction of AChE protein with (**E**) Con A or (**F**) LCA was assayed by Western blotting of the brain extract (100%) and both the recognized (bound) and non-recognized (UB) fractions. Illustrative Western blots of AChE-T species are shown. To the right, densitometric quantification of the percentage of binding of the 75-kDa immunoreactive band to Con A or LCA. The graphs represent mean ± SEM. * indicates statistically different (*p* values < 0.05) with respect to NDC and + significantly different from sAD. One-way ANOVA with Tukey’s multiple comparisons was applied for (**A**,**F**) and Kruskal–Wallis test with Dunn’s multiple comparisons for (**B**,**E**) results.

**Figure 3 ijms-24-01437-f003:**
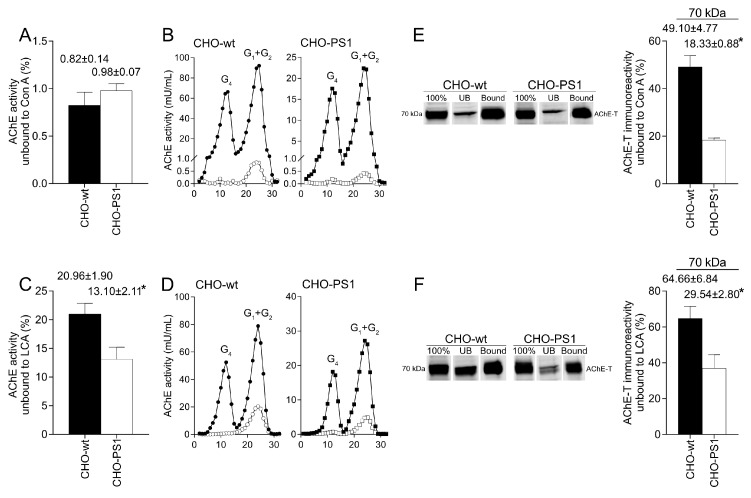
Glycosylation of AChE in CHO cells overexpressing the cholinergic species. Lectin-binding pattern of AChE to *Concanavalina ensiformis* (Con A) and *Lens culinaris* (LCA) lectins in CHO-wt and CHO-PS1. Cells were first transfected with AChE-T+PRiMA encoding plasmids that drive the expression of AChE cholinergic tetramers (see Appendix A). After interaction with lectins, the unbound and bound fractions were separated, and AChE interaction was assessed. AChE enzymatic activity was measured in the fractions unbound to the lectins Con A (**A**) and LCA (**C**), and the percentage of activity with respect to AChE activity of the total extract was calculated. The binding of molecular forms of AChE was also analyzed by ultracentrifugation in sucrose gradients. Representative molecular forms patterns of the total extract (closed symbols) and unbound fraction (open symbols) to Con A (**B**) and LCA (**D**) are shown. The glycosylation of AChE protein was analyzed by Western blotting of the samples prior to interaction with lectins (100%), and also the bound and unbound (UB) fractions. The immunoreactivity of the 70-kDa UB AChE was measured and the percentage with respect to 100% was calculated. Representative Western blots of CHO-wt and CHO-PS1 and percentage of the immunoreactivity UB for Con A (**E**) and LCA (**F**) are shown. Results were confirmed in at least 6 independent determinations from three independent experiments. Data represent the mean ± SEM of the percentage of activity or immunoreactivity over the total prior to interaction. * indicates statistically significant (*p* value < 0.005) when applying unpaired *t*-test.

**Figure 4 ijms-24-01437-f004:**
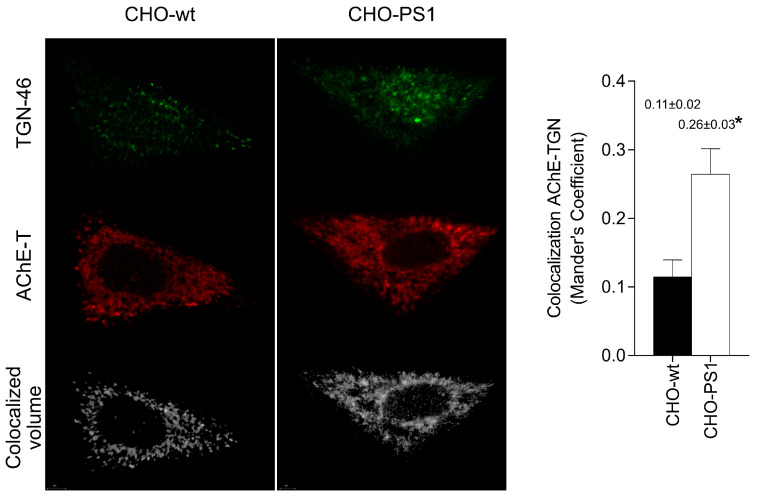
Localization of AChE-T variants in the trans-Golgi network is increased in Chinese hamster ovary cells overexpressing presenilin-1 (CHO-PS1) as compared to wild-type (CHO-wt). CHO cells were transfected with plasmid cDNAs that encode AChE and PRiMA to increase the expression of cholinergic tetrameric AChE. Immunoassay for the trans-Golgi network marker (TGN-46) and AChE-T variant was performed, and images were obtained using a confocal microscope with the ×63 oil immersive objective lens and subsequently analyzed with the Imaris software for colocalization. Representative images of three experiments are shown (CHO-wt, scale bar = 3 μm; CHO-PS1, scale bar = 2 μm). At the bottom, co-localization volume showing pixels positive for both AChE-T and TGN46 marked in white. To the right, quantification of Mander’s coefficient of colocalization. Values are means ± SEM from at least of 12 independent determinations from two independent experiments. * indicates exact *p* value < 0.05 when applying Mann–Whitney test.

**Figure 5 ijms-24-01437-f005:**
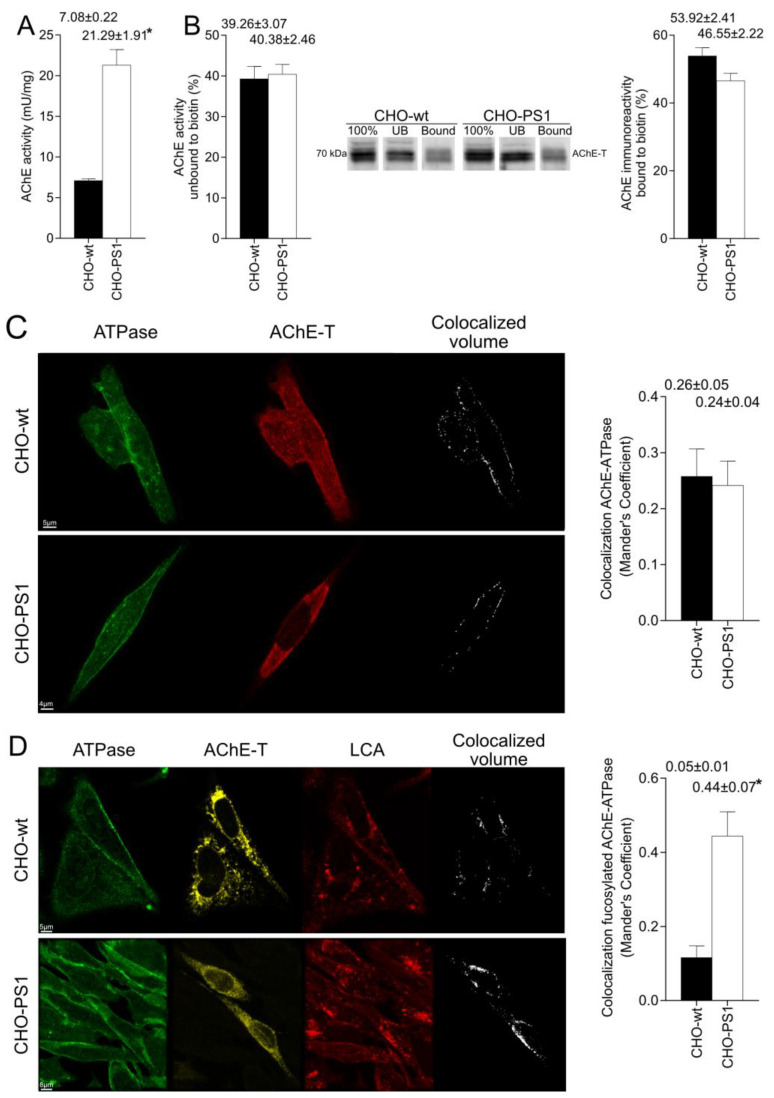
AChE activity and protein in plasmatic membrane of Chinese hamster ovary cells that overexpress PS1 (CHO-PS1) and wild-type (CHO-wt). CHO cells were transfected with AChE and PRiMA1 cDNAs to ensure expression of cholinergic AChE. After 48 h, AChE at the plasmatic membrane was analyzed. (**A**) AChE enzymatic activity was measured at the surface of the cell cultures by a modified Ellman method and normalized to membrane protein. n = 6 independent determinations from three independent experiments. (**B**) Biotinylation of cell surface proteins was assayed (n = 4 samples from each cell type of three independent experiments). Membrane proteins were labelled to biotin, and, after two hours, biotinylated proteins were separated from non-surface protein by interaction with NeutrAvidin. AChE enzymatic activity was measured in the fraction unbound to biotin and expressed as a percentage with respect to the total AChE of non-biotinylated samples (left panel). Biotinylation of AChE protein was analyzed by Western blotting of the unbound and bound fractions to biotin with respect to unbiotinylated sample. A representative Western blot is shown. To the right, densitometric quantification of AChE 70-kDa immunoreactive band in the fraction of membrane proteins labelled to biotin. (**C**) Localization of AChE in plasmatic membrane was achieved by immunoassay for the anti-Na+/K+ ATPase as a membrane marker and AChE-T variant. Images were obtained with confocal microscope using ×63 oil immersive objective lens and colocalization was analyzed afterward with the Imaris software. Representative images of n = 3 experiments are shown. (CHO-wt, scale bar = 5 μm; CHO-PS1, scale bar = 4 μm). To the right, bar graph of Mander’s coefficient of colocalization for AChE-T and ATPase. (**D**) Localization in the membrane of AChE with fucosylated core oligosaccharides. Immunoassay for the lectin Lens culinaris (LCA) using a DyLight^®^ lectin, and antibodies anti-AChE and anti-anti-Na+/K+ ATPase were completed. Confocal images were analyzed with the definition of fucosylated AChE as the co-localized fluorescence for LCA and AChE. Then, colocalization of fucosylated AChE with membrane marker ATPase was calculated. Representative images of n = 3 experiments are shown. (CHO-wt, scale bar = 5 μm; CHO-PS1, scale bar = 8 μm). To the right, Mander’s coefficient for fucosylated AChE and ATPase. The graphs represent mean ± SEM. * indicates *p* value < 0.05 when applying unpaired *t*-test.

## Data Availability

The data presented in the study are included in the article. Further inquiries can be directed to the corresponding authors.

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
