# Peer review of "Presenilin 1 Modulates Acetylcholinesterase Trafficking and Maturation"

_ijms, 2023, doi:10.3390/ijms24021437_

Round 1
Reviewer 1 Report
PS1 can regulate the maturation and trafficking of AChE. Defective regulation could have physio-pathological implications in fAD caused by mutations in PS1, resulting in altered intracellular transport of AChE, which has also been suggested for other key proteins such as TREM2. In sAD, a multifactorial pathology, the changes in AChE glycosylation are more complex to decipher and could also reflect alterations in molecular form patterns, with a specific decrease in tetrameric forms.
The problem addressed in this article is very complex, and this is what the authors state: Further studies are needed to decipher the role of PS1 as a potential chaperone of the cellular location of other glycoproteins. Authors should discuss the results and how they can be interpreted from the perspective of previous studies and of the working hypotheses. The findings and their implications should be discussed in the broadest context possible. Future research directions may also be highlighted.
Although the results are not definitive, the complexity of the therapeutic solution is supported and this fact may trigger research on the biological objective of this article. For this reason, I consider that it is necessary to give it a margin of confidence and be admitted for publication.
Author Response
The problem addressed in this article is very complex, and this is what the authors state: Further studies are needed to decipher the role of PS1 as a potential chaperone of the cellular location of other glycoproteins. Authors should discuss the results and how they can be interpreted from the perspective of previous studies and of the working hypotheses. The findings and their implications should be discussed in the broadest context possible. Future research directions may also be highlighted.
Response: As the reviewer suggest the discussion has been modified in the section that we discuss the potential role of PS1 in the trafficking of the proteins. We have included a recent manuscript that demonstrates the role of PS1 in the location and maturation of the protein with the β-secretase activity, BACE1. This manuscript supports our hypothesis about PS1 regulating the travel of proteins in reticulum and Golgi. Moreover, we have interpreted our results in fAD brains in the context of the role of PS1 as chaperone. Along the discussion we have pointed our results in the perspective of previous results of our group or other published manuscripts.

Reviewer 2 Report
Summary: AChE activity in AD brains is less than that in normal brains. Western blots show that the relative amount of AChE protein in AD and normal brains is the same. The hypothesis was tested that glycosylation differences could explain why AD brains have relatively more inactive AChE protein. Glycosylation levels, measured by binding to lectins, confirmed the hypothesis that AChE protein in AD brains has a different level of glycans whose structures terminate in mannose. Co-expression of the presenilin 1 protein with AChE and PRiMA increased the amount of AChE activity localized on the cell membrane.
Minor comments:
1) Lines 382 – 385 and lines 545 - 549 are comments from a co-author suggesting how to present the results. These comments are not intended for publication.
2) The last sentence in the abstract is confusing. The data show that PS1 increases the level of AChE activity anchored in the cell membrane. It is not clear why an increase in AChE activity in the cell membrane suggests a role for PS1 in the decreased AChE activity in the AD brain.
Author Response
Minor comments:
- Lines 382 – 385 and lines 545 - 549 are comments from a co-author suggesting how to present the results. These comments are not intended for publication.
Many thanks for this indication. We forgot to the suppress the indications for the authors in these sections of the manuscript. These sentences have been deleted in the new version of the manuscript.
2) The last sentence in the abstract is confusing. The data show that PS1 increases the level of AChE activity anchored in the cell membrane. It is not clear why an increase in AChE activity in the cell membrane suggests a role for PS1 in the decreased AChE activity in the AD brain.
As the reviewer indicates, this last sentence is a bit confusing. Our results indicate that PS1 modulates AChE activity in the membrane, however in AD brain there is a reduction in AChE activity. We could not discard that PS1 affect AChE activity in AD, although this effect would be hidden by the greater loss of cholinergic innervation that occurs in AD brain that leads to the reduction on cholinergic elements as acetylcholinesterase, choline-acetyltransferase and indeed of the neurotransmitter acetylcholine. Thus, since this sentence is not important for the conclusions of the manuscript, that should be short and concise, we have decided to suppress this last paragraph in the abstract.
